# Primary Care Professionals’ Self-Efficacy Surrounding Advance Care Planning and Its Link to Sociodemographics, Background and Perceptions: A Cross-Sectional Study

**DOI:** 10.3390/ijerph18179034

**Published:** 2021-08-27

**Authors:** Cristina Lasmarías, Amor Aradilla-Herrero, Cristina Esquinas, Sebastià Santaeugènia, Francisco Cegri, Esther Limón, Mireia Subirana-Casacuberta

**Affiliations:** 1Department of Education and Training, Catalan Institute of Oncology, 08908 Barcelona, Spain; clasmarias@iconcologia.net; 2Catalonia Chronic Care Research Group, University of Vic-Central University of Catalonia, 08500 Vic, Spain; sebastia.santaeugenia@gencat.cat; 3School of Nursing, Escoles Universitàries Gimbernat (Universitat Autònoma de Barcelona), 08174 Sant Cugat del Vallès, Spain; 4Department of Pneumology, Hospital Universitari Vall d’Hebron, 08031 Barcelona, Spain; crise4@hotmail.com; 5Public Health, Mental, Maternal and Child Health Nursing Departament, Faculty of Medicine and Health Sciences, University of Barcelona, 08031 Barcelona, Spain; 6Chronic Care Program, Health Department, Generalitat de Catalunya, 08028 Barcelona, Spain; 7Responsible for Strategy and Innovation of the Association of Family and Community Nursing of Catalonia (AIFiCC), 08001 Barcelona, Spain; pacocegri@gmail.com; 8Primary Care Centre Sant Martí de Provençals, Institut Català de la Salut, 08001 Barcelona, Spain; 9Permanent Board of the Catalan Society of Family and Community Medicine (CAMFiC), 08019 Barcelona, Spain; elimonramirez@gmail.com; 10Primary Care Centre Mataró-7, Institut Català de la Salut, 08019 Mataró, Spain; 11Nursing Department, Parc Taulí Hospital Universitari, Institut d’Investigació i Innovació Parc TaulíI3PT, Universitat Autònoma de Barcelona, Consorci Sanitari Parc Taulí, 08208 Sabadell, Spain; msubiranac@tauli.cat; 12Research Group on Methodology, Methods, Models, and Health and Social Outcomes, Faculty of Health Science and Welfare, Centre for Health and Social Care Research, University of Vic-Central University of Catalonia, 08500 Vic, Spain

**Keywords:** advance care planning, primary care, self-efficacy

## Abstract

Primary care (PC) professionals have been considered the most appropriate practitioners for leading Advance care planning (ACP) processes with advanced chronic patients. Aim: To explore how PC doctors’ and nurses’ self-efficacy surrounding ACP is linked to their sociodemographic characteristics, background and perceptions of ACP practices. Methods: A cross-sectional study was performed. Sociodemographics, background and perceptions about ACP in practice were collected using an online survey. The Advance Care Planning Self-Efficacy Spanish (ACP-SEs) scale was used for the self-efficacy measurement. Statistical analysis: Bivariate, multivariate and backward stepwise logistic regression analyses were performed to identify variables independently related to a higher score on the ACP-SEs. Results: N = 465 participants, 70.04% doctors, 81.47% female. The participants had a mean age of 46.45 years and 66.16% had spent >15 years in their current practice. The logistic regression model showed that scoring ≤ 75 on the ACP-SEs was related to a higher score on feeling sufficiently trained, having participated in ACP processes, perceiving that ACP facilitates knowledge of preferences and values, and perceiving that ACP improves patients’ quality of life. Conclusion: Professionals with previous background and those who have a positive perception of ACP are more likely to feel able to carry out ACP processes with patients.

## 1. Introduction

Advance care planning (ACP) enables individuals to define goals, values, and preferences for future medical treatment and care, to discuss these matters with their family and care providers, and to record and review preferences as necessary [1]. In recent years, several international initiatives have been developed to improve ACP implementation in clinical practice. Initiatives have explored, for instance, the factors that make ACP processes feasible in daily practice [2], the main benefits for patients and families [3] and the health outcomes expected from an ACP process [4,5]. In this sense, proposals about how to integrate models of ACP into health systems [6,7,8] and how to adapt ACP content to individuals’ health condition [9] and to specific settings [10,11] have been studied. 

A wide range of studies define the professional profile that should lead the ACP process. A multidisciplinary approach seems to be the most appropriate to ensure a broad, multidimensional and individualised ACP process [12]. Specifically, primary care (PC) professionals have been recognized as the most suited to promote ACP processes, since the long-term nature of the relationships between PC professionals and patients is unique compared to other fields within the public health system [13,14,15]. This characteristic might help PC professionals build trust with patients, and it could explain the positive attitudes expressed to ACP process and its value on the improvement of the end-of life process [16]. However, PC professionals express difficulties in initiating ACP processes, such as lack of time and skills (skills that are common in other disciplines [6]) and lack of knowledge about how to choose the best moment to initiate an ACP [17]. These obstacles are especially important in the PC setting, where professionals take care of people with diverse disease trajectories [18], and some differences have been found in the approach of physicians and allied health professionals to ACP process [15].

In Spain, recent initiatives have emerged to promote the development of ACP. For example, the ACP model of Spain’s Catalonia region was established within the framework of the 2011–2015 health plan and started in 2014 [19]. Some key aspects of this project included a consensus of experts about the concept of ACP, a practical guide to ACP for clinical practice, and a training programme, including a 10 h online course and face-to-face workshops lasting from 4 to 8 h [19]. In addition, a dedicated section for documenting the ACP process was added to the medical record of complex chronic patients and advanced chronic patients.

In Catalonia, as in other places, advanced chronic patients are considered to benefit when the ACP process begins early [20,21]. These patients are coded as advanced chronic patients using the NECPAL^©^(Palliative Needs) tool (Chair of Palliative Care, Vic, Spain, 2012), which includes the question, “Would I be surprised if this patient died in the next twelve months?” [22,23]. PC professionals are responsible both for coding these patients and recording their ACP process. Documentation of the process is visible to all health professionals in the public health system through a shared clinical record, which is accessible throughout the Catalan public health system [24]. It is understood that having a specific section in the medical record for documenting the ACP is a quality indicator of the implementation of this process [1]. Considering the importance of recording the ACP process, the Catalan model of ACP defined the ACP as a recurring continuity improvement cycle, divided into six phases. One phase was to record preferences and other medical decisions explored in an ACP process as part of the medical record [19]. However, the low quality and accuracy of this documentation in Catalonia are currently areas for improvement, especially considering the ACP record’s potential impact on patient safety [25]. 

Self-efficacy refers to the individual’s belief in his/her ability to carry out a particular task [26]. This topic has frequently been recommended as an indicator for measuring the impact of ACP programs [1,27] and measuring how training programmes improve self-efficacy in ACP [28]. Bandura stated that a person is more self-efficacious when the activity to be carried out (or the learning to be integrated) makes sense and the person also feels prepared to carry it out [26]. In a previous work, the Advance Care Planning Self-Efficacy Spanish scale (ACP-SEs) has been validated to evaluate the self-efficacy of PC, palliative specialists and geriatricians surrounding ACP [29]. These professionals are frequently responsible for carrying out ACP processes with advanced chronic patients. The ACP-SEs scale is composed of 19 items and shows adequate psychometric properties (Cronbach’s alpha = 0.95) to be used with these professionals [29]. Having a validated scale to measure self-efficacy in ACP will facilitate an understanding of this item in our context in order to design and develop public health programs of ACP.

Several projects have been carried out to evaluate the impact of ACP programmes in PC settings with good results [20,30]. However, to our knowledge, the link between, on the one hand, self-efficacy surrounding ACP and, on the other hand, the demographic characteristics, background, and perception of PC doctors and nurses has not been described. Understanding the relationships among these variables in PC in a public health system could help not only the development of public ACP programs but also the integration of ACP in models of care for people with advanced chronic illness. 

We aim to explore the relationship between, on the one hand, PC doctors’ and nurses’ self-efficacy in ACP and, on the other, their sociodemographic characteristics, training and experience, and perception of ACP and to analyse differences between the two disciplines.

## 2. Materials and Methods

### 2.1. Study Design and Selection of Study Subjects

A cross-sectional study was performed. The sample was composed of doctors and nurses from the Catalan society of family and community medicine (CAMFiC) and the Association of family and community nursing of Catalonia (AIFiCC), which represents a large amount of doctors and nurses that work in primary care in Catalonia.

### 2.2. Procedure and Measurements

We created a survey on the *RedCap* platform (https://www.project-redcap.org/, accessed on 12 November 2020). All members of the two scientific societies were invited to participate via a link sent by the presidents of both societies in October 2018. We did not have access to the email addresses of participants. By responding to the survey, participants demonstrated their consent to take part in the study, which was voluntary and anonymous. Three reminders were sent (final reminder in December 2018). The estimated time required to respond to the survey was between 15 and 20 min. 

### 2.3. Measurement Instruments 

The survey was composed of four sections: (1) sociodemographic variables: age, gender, profession and years in current practice; (2) 9 variables about background: knowledge about ACP, previous training in ACP and practical experience in conducting ACP processes; (3) 12 items were created ad-hoc to measure perception, including applicability, of ACP practices (scored from a minimum of 1 to a maximum of 10); (4) the ACP-SEs scale, in its Spanish version [29], containing 19 items rated on a 5-point Likert-type scale (1 = not at all capable and 5 = completely capable). 

### 2.4. Institutional Review Board Statement

All research procedures used in this study were established in accordance with the Declaration of Helsinki. The ethics committee at the University of Vic reviewed and approved the study protocol (code RS005_S). Moreover, we designed the study in accordance with the ethics criteria established by Spanish Organic Law 3/2018 of 5 December on personal data protection and the guarantee of digital rights, following the General Regulation (EU) 2016/679 of 27 April 2016 on data protection. 

### 2.5. Statistical Analysis

Categorical variables were described with absolute frequencies and percentages. Quantitative variables were described using the mean and standard deviation (SD). The Kolmogorov-Smirnov test was used to assess the normality of distributions. 

We analysed the variables related to sociodemographics, background and perception according to field (medicine vs. nursing). The total score on the ACP-SEs was calculated as the sum of scores for the items, which was rescaled from 0 (minimum) to 100 (maximum). In the case of quantitative variables, the Student’s *t*-test (Mann-Whitney U-test if normality was not assumed) or ANOVA tests (in the case of variables with more than two categories) were carried out. The Chi-squared test (Fisher test for frequencies < 5) was used for the comparison of categorical variables. 

To identify the variables related to a high score on the ACP-SEs, we transformed the total score of the scale into a binary variable using the third quartile as a cut-off point (≤ and >75 points). For the bivariate analysis, the variables related to sociodemographics and ACP background were included. From the group about perceptions’ variables, eight were also included: we also transformed these quantitative items into binary variables (< and ≥8 points/10) and included those with a score ≥8 points. 

In the multivariate analysis, we performed a backward stepwise logistic regression analysis to identify variables independently related to ACP > 75 points. Variables with a *p*-value < 0.2 in the bivariate analysis were included as independent factors. The results were described with odds ratios (OR), 95% confidence interval (CI) and *p*-values. The combination of predictors from the final model was used to calculate the probabilities of ACP-SEs > 75 points. A receiver operating characteristic (ROC) analysis and Hosmer-Lemeshow goodness-of-fit test were performed to assess the overall fit of the model [31]. For all the tests, *p*-values < 0.05 were considered statistically significant. We used the statistics package R Studio (V2.5.1) (The R Project for Statistical Computing, Viena, Austria), for the analysis.

## 3. Results

A total of 465 professionals participated in the study, of whom 70.04% were doctors and 29.96% were nurses (one social worker was excluded from the analysis based on field). The mean age was 46.45 years, 81.47% were women, and 66.16% had more than 15 years of professional experience (Table 1). Table 2 shows the results for ACP background (knowledge, training, and experience in ACP). A total percentage of 70.26% had completed training in the subject, of which 30.58% (*n* = 100) had completed more than 8 h. A sum of 52.89% of the participants had carried out an ACP with patients, of which 30.39% (*n* = 141) stated that they had experienced difficulties. The professionals frequently carried out ACPs with people with advanced chronic disease (71.02%), advanced cancer (64.49%) and frailty (62.04%).

The mean score for the item “Do you consider yourself to be sufficiently trained to carry out ACP processes?” was 5.54 out of 10.

Table 3 describes the professionals’ perception of ACP in practice according to field. Of the 12 items explored, 8 show a mean greater than 8 on a scale from 1 to 10. In the set of overall means, the item with the highest score, with a mean of 9.08 (SD = 1.19), is “The ACP process facilitates the expression of wishes and preferences to be taken into consideration when the patient is not able to express by him or herself” and is followed by “ACP is important for complex chronic patients and advanced chronic patients”, with a mean of 8.99 (SD = 1.19). The two items with the lowest score are “ACP is important for healthy people” (mean = 6.77; SD = 2.47) and “The ACP process is feasible in my professional setting” (mean = 7.11; SD = 2.17).

Table 4 shows the results in relation to the 19 ACP-SEs items by field. The ACP-SEs scale shows an overall mean of 65.90 (SD = 16.01) out of 100, with no statistically significant differences between doctors and nurses.

Of the 19 items, 7 show significant differences between doctors and nurses (6, 7, 8, 14, 15, 16, 17, 19). Doctors scored higher on all of these except item 19.

Table 5 shows the bivariate analysis of the main variables of Table 1 and Table 2 and the high-scoring variables from Table 3 (≥8 points), categorised as binary variables, and the score obtained in the ACP-SEs with the cut-off point of the third quartile (≤ and >75 points). A total of 24.52% (*n* = 114) of the participants scored >75 points out of 100. Years in current practice, hours of training in ACP, having previously participated in ACP processes, and believing that ACP offers greater knowledge of patients’ values and preferences, among others, showed statistically significant differences (Table 5).

The result of the logistic regression analysis shows four variables independently related to the increase in the probability of scoring >75 points on the ACP-SEs: having previously participated in ACP processes (OR = 1.70; *p*-value = 0.043), perceiving that ACP contributes to improving people’s quality of life (OR = 1.93; *p*-value = 0.013), perceiving that ACP facilitates the knowledge about patients’ values and preferences (OR = 2.24; *p*-value = 0.028) and considering oneself to be sufficiently trained in ACP (OR = 3.98; *p*-value < 0.001) (Appendix A).

The probabilities for scoring above 75 on the ACP-SEs were obtained by the following formula: Exp(β)/(1 + Exp(β)), where β = −2.838 + 1381 (in case of consider yourself to be sufficiently trained to carry out ACP processes ≥8 points) + 0.528 (previous participation in an ACP process with a patient) + 0.656 (ACP process contributes to improving the patients’ quality of life >8 points) + 0.807 (ACP facilitates knowledge of patients’ values and preferences >8 points). The probability of having >75 points on the ACP-SEs increased with the number of predictors, from 5% when no factor was present to 68% for patients having all four variables (Appendix A). The model is well calibrated with a Hosmer-Lemeshow *p* = 0.875. The predictive power of the final model was AUC = 0.739 ((0.688–0.79)) (Figure 1).

## 4. Discussion

This study describes the relationship between, on the one hand, PC doctors’ and nurses’ self-efficacy in ACP and, on the other, variables related to their sociodemographics, background and perception of ACP. We have shown that, for the most part, professionals are familiar with ACP, have received prior training, and positively value the ACP process for use with people with advanced chronic diseases. Additionally, the factors related to greater self-efficacy in ACP are associated with considering oneself to be sufficiently trained in ACP, having previously participated in ACP processes, believing that ACP can contribute to improving the quality of life of the people cared for and believing that ACP makes it easier to know patients’ values and preferences.

Although we observed no statistical differences across fields in the overall data of the ACP-SEs, there are significant differences in certain items of the scale. These differences are related to the fact that certain aspects explored by the ACP-SEs have to do with medical processes in which nurses are less comfortable to participate, coinciding with findings from international settings [32]. However, nurses have the competencies and skills to lead ACP processes [16,33] and to be able to talk with patients about all aspects of the ACP. Conversations about ACP should focus on promoting people’s autonomy and participation in shared decision-making about their health, their illness, their concerns and preferences, and their values [34] and not exclusively about medical decisions. This holistic conception of the ACP could contribute to strengthen the involvement of all health fields in the process.

One of the most important aspects of our study is that training and implementation of ACP processes as part of the clinical routine are linked to greater self-efficacy in ACP. In this sense, training can improve self-efficacy in ACP [28,35], because it is key to promoting the development of safe practices in professionals who feel insecure or describe barriers when carrying out these processes.

Although, to our knowledge, there are no standard training models, specific short training sessions can stimulate the implementation of ACP processes among professionals [36], which is consistent with our results. The self-efficacy measurement can be useful for exploring the design and impact of training programmes [27] and recognising gold standard professionals who have a greater predisposition to lead ACP processes. This will allow the identification of ACP experts or facilitators within teams, especially in nursing [37].

In light of the data from our study, we suggest that ACP training include awareness of the importance of the ACP process for patients, reflections about barriers in practice and how to manage them, and the specification of aspects to be discussed with the sick person. The content of the training should promote a multidisciplinary approach that integrates all the dimensions of the individual, beyond discussion about medical treatments [33]. Finally, it could be useful to use simulation practices and reflective diaries about daily practice, which facilitate a learning process focussed on the professional’s real environment. Education about ACP should proceed gradually, training professionals in how to discuss ACP with patients firstly with a low degree of complexity and later with patients with more complex needs, according to the experience of the professional. This would allow the professional responsible for driving the ACP to gain confidence and greater self-efficacy in facing more complex ACP processes that necessitate wide-ranging reflection. 

Another result to be highlighted is related to the feasibility of implementing ACP, since this category received one of the lowest scores. Further analysis could examine the specific relationships between feasibility and other variables. Previous studies have shown that one of the barriers to ACP is the lack of time and the unavailability of spaces that facilitate respectful dialogue [14]. Policymakers and administrators who develop ACP programmes should consider not only the motivation and training of professionals, but also the improvement of ACP processes, their incorporation into team planning, the creation of meeting spaces to carry out the process, the development of incentives for professionals and the recording of the ACP process in medical records in a way that is both accessible and visible [25]. 

Finally, our participants gave high scores to the importance of ACP in patients with advanced and complex diseases, mainly cancer, organ disease, and frailty. Although a PC professional is best positioned to conduct the PCA process [38], given the high prevalence of people with chronic diseases and palliative care needs in the community [22], future studies should analyse facilitators and barriers to the implementation of ACP within an integrated health system [18,20], including the participation of professionals from multiple disciplines and different settings (acute care hospitals, intermediate care hospitals, mental health facilities and palliative home care teams). Such an approach would facilitate decision-making in patients with diverse trajectories and diseases.

ACP has the potential to promote shared decision-making and it is therefore essential that professionals feel prepared and motivated to lead the process. After the health threat caused by the SARS-COVID-19 pandemic, it is especially crucial that institutions, healthcare professionals in general and PC in particular, strengthen their efforts to improve healthcare quality by continuing to integrate practices of ACP in their daily routine [4,39].

### 4.1. Strengths

The results are based on a large sample of doctors and nurses from all over Catalonia who care for people with advanced chronic diseases and who work in a public health system that adopted an ACP process more than 5 years ago. Another strength is that we included both doctors and nurses and broke down the results by field, allowing us to see differences in the link between ACP self-efficacy and the other variables across the two fields. Finally, we used a validated self-efficacy scale, improving the accuracy of the results.

### 4.2. Limitations of Study

Since participation in the study was voluntary, the survey responses may be biased by participants’ motivation towards and interest in the subject. This could explain the high scores in most of the survey items. Additionally, because only PC professionals were surveyed, we must be cautious when extrapolating the data to other healthcare settings. Such settings could be examined in future studies. The variables related to ACP were measured with a numerical scale that was not accompanied by qualitative descriptors, making it somewhat difficult to interpret participants’ responses. The cut-off point of 8 was decided by the team and may have been influenced by the team members’ own interpretation of the numerical scale. Finally, the data obtained cannot be extrapolated or applied to healthcare systems that are very different from the Catalan system.

### 4.3. Implications for Practice

The data from this study can contribute to improving the design and implementation of ACP programmes in PC settings, especially in relation to the definition of possible competencies and levels of responsibility in the implementation of ACP processes by the professionals involved. The study makes it possible to identify professionals that are most suited to act as facilitators of ACP processes, especially for complex cases.

## 5. Conclusions

To our knowledge, this is the first study that analyses in detail the relationship between self-efficacy and the perception of PC professionals about ACP practices in the context of an integrated ACP model within a public health system. We have identified that considering oneself to be sufficiently trained in ACP, having previously participated in ACP processes, believing that ACP can contribute to improving the quality of life of the people cared for and believing that ACP makes it easier to know patients’ values and preferences are factors linked to PC professionals feeling more able to carry out ACP processes with patients. Strategies to improve the feasibility of ACP in practice could include systematically integrating ACP into care models, improving recording systems in medical records, and identifying and promoting predictive and facilitating factors for the implementation of ACP in care for people with advanced chronic diseases. This article provides important information about the role of self-efficacy in ACP practices in interdisciplinary PC teams, and it may help policy makers and administrators to promote the ACP process within public health systems.

## Figures and Tables

**Figure 1 ijerph-18-09034-f001:**
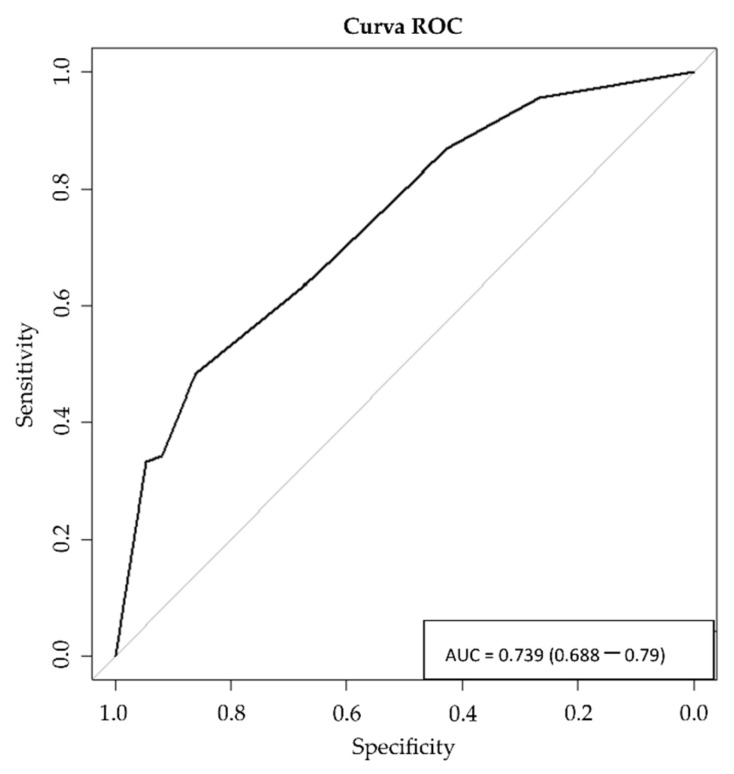
ROC curve ^1^. Predictive value of the model. AUC = 0.739 (0.688–0.79) ^2^. (^1^ ROC = Receiver operating characteristic curve; ^2^ AUC = Area Under Curve).

**Table 1 ijerph-18-09034-t001:** Sociodemographic characteristics and field.

		Total *n* = 464 *	Medicine *n* = 325	Nursing *n* = 139
Age	Mean (SD)	46.45 (10.17)	46.38 (10.23)	46.61 (10.04)
Gender	Female	379 (81.47%)	253 (77.85%)	125 (89.93%)
	Male	86 (18.53%)	72 (22.15%)	14 (10.07%)
Years in current practice	<1 year	1 (0.22%)	1 (0.31%)	0
	1–5 years	60 (12.93%)	47 (14.46%)	13 (9.35%)
	6–10 years	45 (9.7%)	35 (10.77%)	10 (7.19%)
	11–15 years	45 (9.7%)	34 (10.46%)	11 (7.91%)
	>15 years	307 (66.16%)	204 (62.77%)	103 (74.1%)
	No active	6 (1.29%)	4 (1.23%)	2 (1.44%)

* The social worker has been excluded.

**Table 2 ijerph-18-09034-t002:** Training and professional experience in ACP ^1^ according to field.

		Total *n* = 464	Medicine *n* = 325	Nursing *n* = 139
Have you heard of ACP?	No	12 (2.59%)	9 (2.77%)	3 (2.16%)
	Yes	452 (97.41%)	316 (97.23%)	136 (97.84%)
Have you completed training in ACP?	No	138 (29.74%)	108 (33.23%)	30 (21.58%)
	Yes	326 (70.26%)	217 (66.77%)	109 (78.42%)
Number of hours of training completed (*n* = 326)	<1 h	9 (2,75%)	7 (3.23%)	2 (1.83%)
	1 h–2 h	66 (26.61%)	50(23.04%)	16 (18.35%)
	2 h–4 h	87 (19.88%)	55 (25.35%)	32 (14.68%)
	4 h–8 h	64 (20.18%)	44 (20.28%)	20 (29.36%)
	>8 h	100(30.58%)	61 (28.11%)	39 (35.78%)
Do you consider yourself to be sufficiently trained to carry out ACP processes?	Value: 1 to 10 $\bar{x}$ (SD)	5.54 (2.29)	5.44 (2.26)	5.78 (2.37)
Have you participated in an ACP process with a patient?	No	219 (47.2%)	149 (45.85%)	70 (50.36%)
	Yes	245 (52.89%)	176 (54.15%)	69 (49.64%)
Number of ACP processes per month (*n* = 245)	None	81 (33.06%)	59 (33.52%)	22 (31.88%)
	1–5	155 (63.27%)	112 (63.64%)	43 (62.32%)
	6–10	6 (2.45%)	3 (1.7%)	3 (4.35%)
	11–20	3 (1.22%)	2 (1.14%)	1 (1.45%)
Time since the last ACP process	Less than one week	34 (13.88%)	19 (10.8%)	15 (21.74%)
	Between one week and one month	70 (28.57%)	54 (30.68%)	16 (23.19%)
	More than one month	141 (57.55%)	103 (58.52%)	38 (55.07%)
Have you had any difficulties in carrying out ACP processes?	No	104 (22.41%)	75 (23.08%)	29 (20.86%)
	Yes	141 (30.39%)	101 (31.08%)	40 (28.78%)
Main disease of the patients with whom you carried out ACP processes (more than one option was possible)	Advanced organ failure disease	174 (71.02%)	126 (71.59%)	48 (69.57%)
	Advanced cancer	158 (64.49%)	116 (65.91%)	42 (60.87%)
	Advanced dementia	99 (40.41%)	67 (38.07%)	32 (46.38%)
	Advanced neurological disease	77 (31.43%)	53 (30.11%)	24 (34.78%)
	Frailty	152 (62.04%)	108 (61.36%)	44 (63.77%)
	Another chronic disease	88 (35.92%)	65 (36.93%)	23 (33.33%)

^1^ ACP = advance care planning.

**Table 3 ijerph-18-09034-t003:** Perception of ACP ^1^ practices by field.

	Total *n* = 464 (Mean, SD)	Medicine *n* = 325 (Mean, SD)	Nurse *n* = 139 (Mean, SD)
The ACP process facilitates the expression of wishes and preferences to take into account when the patient is not able to express him or herself	9.08 (1.19)	9.01 (1.22)	9.26 (1.1)
ACP is important for complex chronic patients and advanced chronic patients	8.99 (1.19)	8.91 (1.21)	9.18 (1.14)
ACP facilitates knowledge of patients’ values and preference	8.98 (1.11)	8.91 (1.12)	9.13 (1.08)
ACP makes it possible to identify the patient’s personal representative	8.78 (1.46)	8.76 (1.44)	8.82 (1.5)
ACP enables the patient to die at the place he/she wishes	8.40 (1.66)	8.33 (1.63)	8.58 (1.72)
ACP process makes it possible to adapt treatments to realistic therapeutic options	8.26 (1.58)	8.15 (1.57)	8.5 (1.59)
ACP is important for patients with a chronic disease even they are not identified as complex chronic patients or advanced chronic patients	8.24 (1.71)	8.23 (1.68)	8.28 (1.79)
The ACP process contributes to improving patients’ quality of life	8.12 (1.71)	8.05 (1.7)	8.27 (1.71)
ACP helps me to coordinate with other professionals	7.97 (1.85)	7.78 (1.92)	8.41 (1.61)
ACP gives me confidence as a professional that I’m caring for patients properly	7.92 (1.88)	7.9 (1.85)	7.96 (1.96)
The ACP process is feasible in my professional setting	7.11 (2.17)	6.95 (2.23)	7.46 (2.01)
ACP is important to healthy people	6.77 (2.47)	6.74 (2.37)	6.83 (2.69)

^1^ ACP = advance care planning; Correlation between the ACP-SEs and the Groups of Interest.

**Table 4 ijerph-18-09034-t004:** Relation between the ACP-SEs ^1^ and field.

	Total *n* = 464 (Mean, SD)	Medicine *n* = 325 (Mean, SD)	Nurse *n* = 139 (Mean, SD)	*p*-Value
1. Find the time to discuss the patient’s prognosis preferences and care plan with the patient	3.48 (0.98)	3.49 (0.98)	3.45 (0.97)	0.60
2. Determine how much the patient wants to know about the prognosis	3.68 (0.82)	3.73 (0.78)	3.58 (0.89)	0.10
3. Determine the level of involvement the patient wants in decision-making	3.65 (0.81)	3.69 (0.78)	3.54 (0.86)	0.07
4. Determine who else (e.g., family members) the patient would like to be involved in decision-making	3.98 (0.75)	4.03 (0.73)	3.88 (0.8)	0.07
5. Provide the desired level of information and guidance needed to help the patient in decision-making	3.69 (0.81)	3.71 (0.77)	3.64 (0.88)	0.40
6. Describe the pros and cons of different life-sustaining treatments	3.5 (0.92)	3.56 (0.87)	3.37 (1.02)	**0.049**
7. Determine the patient’s specific wishes for types of medical treatment	3.53 (0.85)	3.58 (0.84)	3.41 (0.86)	**0.049**
8. Discuss and negotiate individualised treatment goals and plans with patient	3.55 (0.89)	3.6 (0.86)	3.42 (0.93)	**0.049**
9. Ensure that patient’s treatment preferences will be honoured at your facility	3.98 (0.83)	4.01 (0.81)	3.93 (0.87)	0.40
10. Ensure that patient’s treatment preferences will be honoured at a hospital if patient is hospitalised	2.77 (1.13)	2.73 (1.1)	2.87 (1.2)	0.20
11. Discuss how to complete a living will with the patient	3.5 (1.1)	3.46 (1.11)	3.6 (1.09)	0.20
12. Determine when there should be a shift in care goals	3.4 (0.98)	3.36 (0.96)	3.5 (1.04)	0.20
13. Reassess the patient’s wishes when a shift in care goals is needed	3.59 (0.95)	3.57 (0.96)	3.65 (0.92)	0.40
14. Openly discuss uncertainty with patient when it exists	3.81 (0.85)	3.87 (0.82)	3.67 (0.9)	**0.02**
15. Educate patient and clarify any misperceptions about the disease or prognosis	3.8 (0.77)	3.86 (0.72)	3.63 (0.86)	**0.006**
16. Respond empathetically to patient’s and family’s concerns	4.11 (0.72)	4.18 (0.67)	3.95 (0.81)	**0.003**
17. Communicate “bad news” to patients and their families	3.67 (0.84)	3.84 (0.71)	3.29 (0.99)	**<0.001**
18. Engage patients in ACP conversations	3.69 (0.88)	3.71 (0.82)	3.64 (1.01)	0.50
19. Correctly register the decisions and care plan agreed to over the course of the ACP	3.69 (1.06)	3.62 (1.08)	3.83 (1.01)	**0.049**
TOTAL SCORE Mean (SD) (re-scaled to 100)	65.90 (16.01)	66.59 (15.40)	64.28 (17.30)	0.200

^1^ ACP-SEs = Advance Care Planning Self-Efficacy Spanish. *p*-values < 0.05; Statistically significant differences have been marked in bold.

**Table 5 ijerph-18-09034-t005:** Comparison of the total ACP score categorized according to Q3 cut-off of 75 points on the APC-SEs scale and sample characteristics and variables of interest in the questionnaire. Bivariate analysis.

		N = 465 ^1^ (100%)	Total Score ACPS-SEs ^2^ (100 Points)		
			<= 75	>75	*p*-Value
**Age**	**<50 years**	273 (58.71%)	208 (59.26%)	65 (57.02%)	0.754
	≥50 years	192 (41.29%)	143 (40.74%)	49 (42.98%)	
Gender	Female	379 (81.51%)	291 (82.91%)	88 (77.19%)	0.220
	Male	86 (18.49%)	60 (17.09%)	26 (22.81%)	
Field	Medicine	325 (69.89%)	243 (69.43%)	82 (71.93%)	0.698
	Nursing	139 (29.89%)	107 (30.57%)	32 (28.07%)	
Years in current practice	≤15 years	151 (32.90%)	120 (34.38%)	31 (28.18%)	**<0.001**
	>15 years	308 (67.10%)	229 (65.62%)	79 (71.82%)	
Have you completed training in ACP?	No	138 (29.68%)	111 (31.62%)	27 (23.68%)	0.135
	Yes	327 (70.32%)	240 (68.38%)	87 (76.32%)	
Number of hours of training completed	≤4 h	162 (49.54%)	131 (54.58%)	31 (35.63%)	**0.004**
	>4 h	165 (50.46%)	109 (45.42%)	56 (64.37%)	
Do you consider yourself to be sufficiently trained to carry out ACP processes?	<8 points	361 (77.63%)	302 (86.04%)	59 (51.75%)	**0.001**
	≥8 points	104 (22.37%)	49 (13.96%)	55 (48.25%)	
Have you participated in an ACP process with a patient?	No	219 (47.1%)	186 (52.99%)	33 (28.95%)	**<0.001**
	Yes	246 (52.9%)	165 (47.01%)	81 (71.05%)	
Number of ACP processes per month	None	81 (32.93%)	56 (33.94%)	25 (30.86%)	**<0.001**
	≥1	156 (63.42%)	109 (66.06%)	56 (69.14%)	
Have you had any difficult in carrying out ACP processes?	No	105 (22.58%)	56 (33.94%)	49 (60.49%)	**<0.001**
	Yes	141 (30.32%)	109 (66.06%)	32 (39.51%)	
ACP is important for complex chronic patients and advanced chronic patients	≤8 points	135 (29.03%)	119 (33.90%)	16 (14.04%)	**0.003**
	>8 points	330 (70.97%)	232 (66.10%)	98 (85.96%)	
ACP is important for patients with a chronic disease even they are not identified as complex chronic patients or advanced chronic patients	≤8 points	237 (50.97%)	197 (56.13%)	40 (35.09%)	0.632
	>8 points	228 (49.03%)	154 (43.87%)	74 (64.91%)	
The ACP process facilitates expression of wishes and preferences to taken into account when the patient is not able to express him or herself	≤8 points	114 (24.52%)	97 (27.64%)	17 (14.91%)	0.097
	>8 points	351 (75.48%)	254 (72.36%)	97 (85.09%)	
ACP makes it possible to identify the patient’s personal representative	≤8 points	153 (32.90%)	130 (37.04%)	23 (20.18%)	**0.013**
	>8 points	312 (67.10%)	221 (62.96%)	91 (79.82%)	
The ACP process contributes to improving patients’ quality of life	≤8 points	241 (51.83%)	203 (57.83%)	38 (33.36%)	**<0.001**
	>8 points	224 (48.17%)	148 (42.17%)	76 (66.67%)	
ACP enables the patient to die at the place he/she wishes	≤8 points	202 (43.44%)	171 (48.72%)	31 (27.19%)	**<0.001**
	>8 points	263 (56.56%)	180 (51.28%)	83 (72.81%)	
The ACP process makes it possible to adapt the treatments to realistic therapeutic options	≤8 points	120 (25.81%)	97 (27.64%)	23 (20.18%)	**<0.001**
	>8 points	345 (74.19%)	254 (72.36%)	91 (79.82%)	
ACP facilitates knowledge of patients’ values and preferences	≤8 points	46 (9.89%)	42 (11.97%)	4 (3.51%)	**<0.001**
	>8 points	419 (90.11%)	309 (88.03%)	110 (96.49%)	

^1^ Including the social worker. ^2^ ACP-SEs = Advance Care Planning Self Efficacy Spanish *p*-values < 0.05; Statistically significant differences have been marked in bold

## Data Availability

Data sharing not applicable.

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
