# Peer review of "Primary Care Professionals’ Self-Efficacy Surrounding Advance Care Planning and Its Link to Sociodemographics, Background and Perceptions: A Cross-Sectional Study"

_ijerph, 2021, doi:10.3390/ijerph18179034_

Round 1

Reviewer 1 Report

The manuscript reagrding the Primary Care Professionals’ Self-efficacy on Advance Care Planning has been written in a comprehensive way, with an approriate data analysis and interpretation. I have a few minor revisions to suggest to the authors: 
Line 84, the author should state the reference in the sentence “Documentation of the process is visible to all health professionals in the public health system through a shared clinical record, which is accessible throughout the Catalan public health system.”
Page 3, line 146
The author mention that Qualitative variables were described with absolute frequencies and percentages. Did they refer to Categorical variables here?

Page 4, line 172, i would suggest to not start the sentence with the number, insted the authors might consider to use “ A total of 465 professionals…”

Table 2, it is not clear why the responses “No” of the question “Have you heard of ACP”, are in bold. Is this a typing error, or is there a reason behing this reporting?

I woul suggest to delete in Table 1, 2, 3 and 4 the %  from each of the following:
TOTAL N=464 (100%) Medicine n=325 (70.04%) Nursing n=139 (29.96%)
It create confusion, especially in Table 3 and 4, when your report the mean and standard deviation.
In table 3 and 4, the authors might consider to delete from the table description the part “Mean (SD)” and “Mean (SD), p-value, respectively, and include it in the heading of the Table, eg. TOTAL N=464, Mean (SD), Medicine n=325, Mean (SD), Nursing n=139, Mean (SD).

Author Response

The authors would like to thank the reviewer for his/her thorough review of the manuscript. We believe that this revised version, which includes most of the reviewers’ suggestions, is more accurate and communicates better the main message of the article. Nevertheless, we will gratefully respond to any further comments on the text.

In order to facilitate the understanding of the comments, the specific answers will be shown in regular form and the text included or modified in the main manuscript will be shown in italics, if it is necessary.

Following comments have been modified in a new version of the manuscript:

  • Page 4, line 172, I would suggest to not start the sentence with the number, instead the authors might consider to use “ A total of 465 professionals…” (Line 175)
  • Table 2, it is not clear why the responses “No” of the question “Have you heard of ACP”, are in bold. Is this a typing error, or is there a reason behind this reporting? (It was a mistake)
  • I would suggest to delete in Table 1, 2, 3 and 4 the %  from each of the following:
    TOTAL N=464 (100%) Medicine n=325 (70.04%) Nursing n=139 (29.96%).
    It creates confusion, especially in Table 3 and 4, when your report the mean and standard deviation. 
  • In table 3 and 4, the authors might consider to delete from the table description the part “Mean (SD)” and “Mean (SD), p-value, respectively, and include it in the heading of the Table, eg. TOTAL N=464, Mean (SD), Medicine n=325, Mean (SD), Nursing n=139, Mean (SD). (Line 201; Line 207)

Regarding the following two comments, please see the specific answer as follows:

  • Line 84, the author should state the reference in the sentence “Documentation of the process is visible to all health professionals in the public health system through a shared clinical record, which is accessible throughout the Catalan public health system.”

Thanks for your comment. We have added this reference, which explains the Catalan Model of Chronic and integrated care:

Contel JC, Ledesma A, Blay C, et al. Chronic and integrated care in Catalonia. Int J Integr Care. 2015;15:e025. Published 2015 Jun 29. doi:10.5334/ijic.2205. (Line 88)

  • Page 3, line 146. The author mention that Qualitative variables were described with absolute frequencies and percentages. Did they refer to Categorical variables here?

Thanks for your suggestion. We have modified the wording according to your question. (Line 149)

Thank you again for taking the time to review our paper and for your constructive comments.

Reviewer 2 Report

Thank you for giving me the opportunity to review the manuscript, entitled “Primary care professionals’ self-efficacy surrounding advance care planning and its link to sociodemographics, background and perceptions: a cross-sectional study”. Please see below some comments and suggestions. 

Introduction

-Consider briefly discuss primary care profession’s background and perception of advance care planning with support of literature, such as:

Fan, E., & Rhee, J. J. (2017). A self-reported survey on the confidence levels and motivation of New South Wales practice nurses on conducting advance-care planning (ACP) initiatives in the general-practice setting. Australian journal of primary health23(1), 80-86.

Hafid, A., Howard, M., Guenter, D., Elston, D., Fikree, S., Gallagher, E., ... & Waters, H. (2021). Advance care planning conversations in primary care: a quality improvement project using the Serious Illness Care Program. BMC palliative care20(1), 1-10.

Results

- On p.4 line 172, please delete. After 465

- On P.4 line 180, suggest to modify “chronic organ disease” to chronic disease

- On p.5 line 3 196, suggest to review (mean=7.11), given that Table 3 listed mean as 7.10.

- For Table 2 on p.5 row “Number of hours of training completed” , please use dot (instead of comma) to indicate decimal place

- On Table 2  on p.5 row “Main disease for the patients with whom you carried out ACP processes “. Please clarify what is advanced chronic organic disease? How does that differ from another chronic disease?

- On p.6 Table 4 , consider changing title Medicine to Doctor.

- On p.7, item 14 “Openly discuss uncertainty with patient wen it exists”, suggest to review the format and bold 0.02

- On p.9 Table 5 row “ACP facilitates knowledge of patients’ values and preferences”, p value is missing. Please review.

- On p.9 line 287, “…the final model was AUC =0.739 ((0,688-0,79))”. Please review as it is inconsistent with Figure 1 description on p.10 line 242, where it stated “AUC2= 0/739. (0.688-0.79).”

Reference

- For the reference list, please review all and follow the format as listed in https://www.mdpi.com/authors/references

- For instance, please italic the journal’s name. Please use semi-colon between author’s names. Please add full-stop/dots after each author’s initials.

Author Response

The authors would like to thank the reviewer for his/her thorough review of the manuscript. We believe that this revised version, which includes most of the reviewers’ suggestions, is more accurate and communicates better the main message of the article. Nevertheless, we will gratefully respond to any further comments on the text.

In order to facilitate the understanding of the comments, the specific answers will be shown in regular form and the text included or modified in the main manuscript will be shown in italics.

The following suggestions have been modified in a new version of the manuscript:

  • On p.4 line 172, please delete. After 465 (Line 175)
  • On P.4 line 180, suggest to modify “chronic organ disease” to chronic disease (Line 183)
  • On p.5 line 3 196, suggest to review (mean=7.11), given that Table 3 listed mean as 7.10. (Changed on table 3)
  • For Table 2 on p.5 row “Number of hours of training completed” , please use dot (instead of comma) to indicate decimal place.
  • On p.7, item 14 “Openly discuss uncertainty with patient wen it exists”, suggest to review the format and bold 0.02
  • On p.9 line 287, “…the final model was AUC =0.739 ((0,688-0,79))”. Please review as it is inconsistent with Figure 1 description on p.10 line 242, where it stated “AUC2= 0/739. (0.688-0.79).” (Figure changed, line 246)

Regarding the following two comments, please see the specific answer as follows:

  • Consider briefly discuss primary care profession’s background and perception of advance care planning with support of literature, such as:
    • Fan, E., & Rhee, J. J. (2017). A self-reported survey on the confidence levels and motivation of New South Wales practice nurses on conducting advance-care planning (ACP) initiatives in the general-practice setting. Australian journal of primary health23(1), 80-86.
    • Hafid, A., Howard, M., Guenter, D., Elston, D., Fikree, S., Gallagher, E., ... & Waters, H. (2021). Advance care planning conversations in primary care: a quality improvement project using the Serious Illness Care Program. BMC palliative care20(1), 1-10.

Thank you very much for your suggestion. According to that, we have added some comments regarding to both papers:

Line 65-67: and it could explain the  good attitudes expressed to ACP process and its value on the improvement of the end-of life process (16)

Line 71-72: (18), and some differences have been found on physicians and allied health professionals approach to ACP process(15).

  • On Table 2  on p.5 row “Main disease for the patients with whom you carried out ACP processes “. Please clarify what is advanced chronic organic disease? How does that differ from another chronic disease?

Thank you for your question. As we explain in the Introduction section (line 81-84), in Catalonia we code patients as chronic advanced patients using the NECPAL tool. This tool, based on the common illness trajectories, classified patients including organ failure diseases as lung, heart, kidney, and liver. In table 2 we applied this classification. In order to better clarify this question, we have changed the original wording by “advanced organ failure disease” Table 2, row 5.

  • On p.6 Table 4 , consider changing title Medicine to Doctor.

Thanks for your comment. The authors discussed about talking about Doctors and Nurses, but finally we specifically wrote Medicine and Nursing because we are talking about the field, so we think Medicine fits better to the wording.

  • On p.9 Table 5 row “ACP facilitates knowledge of patients’ values and preferences”, p value is missing. Please review.

Thank you for this comment. We think that p-value is shown in table 5. Please, take into consideration that there are two sub-rows and probably this could be confusing. Is this sense, p-value described is:  <0.001

  • References
    • For the reference list, please review all and follow the format as listed in https://www.mdpi.com/authors/references
    • For instance, please italic the journal’s name. Please use semi-colon between author’s names. Please add full-stop/dots after each author’s initials.

We have reviewed and modified the reference list according to the https://www.mdpi.com/authors/references .

Thank you again for taking the time to review our paper and for your constructive comments.